# The paradoxical decline and growth of trust as a function of borderline personality disorder trait count: Using discontinuous growth modelling to examine trust dynamics in response to violation and repair

Gamze Abramov[1]*, Sebastien Miellet[1], Jason Kautz[2¤], Brin F. S. Grenyer[1], Frank P. Deane[1]*

1 School of Psychology, University of Wollongong, Wollongong, NSW, Australia, 2 Darla Moore School of Business, University of South Carolina, Columbia, South Carolina, United States of America

¤ Current address: Naveen Jindal School of Management, University of Texas, Dallas, Texas, United States of America

* ga385@uowmail.edu.au (GA); fdeane@uow.edu.au (FPD)

## Abstract

Borderline personality disorder (BPD) is associated with paradoxical trust cognitions and behaviours. While BPD is associated with difficulty forming trust and maintaining cooperation in trust-based exchanges, design and analytical methodology best suited to reveal the temporal ebb and flow of trust have been underutilized. We used an economic game to examine the trajectories of trust as it forms, dissolves, and restores in response to trust violation and repair, and to explain how these vary as a function of borderline pathology. Young adults ($N = 234$) played a 15-round trust game in which partner trustworthiness was varied to create three phases: trust formation, trust violation, and trust restoration. Discontinuous growth modelling was employed to capture the trends in trust over time and their relationship with BPD trait count. BPD trait count was associated with an incongruous pattern of trust behaviour in the form of declining trust when interacting with a new and cooperative partner, and paradoxically, increasing trust following multiple instances of trust violation by that partner. BPD trait count was also associated with trust restoring at a faster rate than it was originally formed. By adopting a methodology that recognizes the dynamic nature of trust, this study illustrated at a micro level how relational disturbances may be produced and maintained in those with a moderate to high BPD trait count. Further investigation of the factors and processes that underlie these incongruous trust dynamics is recommended.

## Introduction

Borderline Personality Disorder (BPD) is a complex and often enduring disorder with a prevalence rate of 1–2% in the community [1], and 15 to 20% among patients in psychiatric

**Data Availability Statement:** All relevant data are within the manuscript and its Supporting Information files.

**Funding:** This research received financial funding by the University of Wollongong. The funders had no role in study design, data collection and analysis, decision to publish, or preparation of the manuscript.

**Competing interests:** The authors have declared that no competing interests exist.

hospitals or outpatient clinics [2]. Disturbed interpersonal functioning has been identified as a core component of BPD in factor analytic studies [3, 4], and is one of the strongest diagnostic discriminators of the disorder [5]. Long-term prospective studies show that even when symptoms remit, improvement in social functioning is limited, with profound and persistent relational impairments [6, 7]. Social network analysis has revealed that individuals with BPD experience a greater number of conflicted relationships and are typically 'cut off' from more people in their networks [8–10], suggesting their relationships are marred by rupture and, potentially, a lack of reconciliation. Since the maintenance of stable partner relationships is associated with recovery [11], it is essential to identify the relational dynamics that contribute to the instability and breakdown of relationships.

Impairments in the capacity to trust has been proposed as a key factor in understanding the relational disturbances of BPD [12]. Believing that others will betray, exploit and deceive is characteristic of the disorder [13, 14]. A systematic review on early maladaptive schemas found the mistrust/abuse schema was one of the most highly endorsed among people with BPD traits or a diagnosis [15]. This suggests that these individuals perceive the world and others as malevolent, hostile, and dangerous. Accordingly, experimental studies using economic exchange games have found that individuals with BPD often behave in ways that compromise the formation and maintenance of trust, particularly once trust has been ruptured [16–18].

Among non-clinical populations, diminished interpersonal trust within intimate relationships is associated with diminished perceptions of relationship quality [19], and that fluctuate markedly within short periods of time [20]. Moreover, breach of trust is associated with relationship dissatisfaction, and is considered to be a relevant factor when deliberating whether to end a relationship [21]. As such, the way that individuals navigate the vicissitudes in trust is likely to impact the quality, stability, and longevity of relationships, and is therefore relevant to improving interpersonal functioning in BPD. Using an economic game paradigm, the current study seeks to examine how the presence of BPD traits modifies interpersonal trust processes, including how trust changes in response to violation and repair.

## The Trust Game (TG)

The experimental paradigm most frequently used to examine trust behaviours is the trust game [TG: 22]. Typically, in a two-player game, one person—the *investor*—is allocated an endowment (e.g. $10) and can choose whether to entrust any of it with the other player—the *trustee*—for investment. The amount invested is automatically multiplied by a factor—most often three—before being received by the trustee. The trustee can then reciprocate if they so desire, by sending the investor a sum of their choosing from this tripled amount. Trust is operationalized as the proportion of the original endowment transferred by the investor. The TG can be played with a human dyad or a programmed agent, the latter allowing researchers to manipulate responses such as the magnitude of the sum repaid, to investigate how trust changes over time and in response to discrete events.

## Interpersonal trust as a dynamic phenomenon

While the TG is an ideal vehicle for examining trust dynamically, the methodological and data analytical procedures best suited to reveal the dyadic ebb and flow of trust and cooperation across repeated interactions have been underutilized [see 23–25], with a few exceptions within the organizational psychology sphere [e.g., 24, 26].

Trust as a temporal phenomenon has been conceptualized as comprising at least three distinct phases: formation, dissolution, and restoration [18, 24, 27–31]. Trust formation refers to the development of trust in a new relationship; trust dissolution refers to the decline of trust in

response to a violation of one's trust by another party; and trust restoration refers to the rebuilding of trust subsequent to trust dissolution, in response to reparative attempts by the offending party. Given that individuals with BPD behave in ways that jeopardise trust development and maintenance [16–18], it is important to conduct research that captures its temporal and mutable nature.

## Trust and BPD

In the BPD literature, trust has not typically been treated as a dynamic and multiphasic construct. Rather, it has been measured at a single point in time [32], aggregated across multiple time points [18, 33], or restricted to focus on a single phase of trust [17]. To the authors' knowledge, only one study has accounted for trust dissolution and trust restoration by manipulating trustee reciprocity to differentiate between three phases: cooperative reciprocity, where the trustee returned a profit across five consecutive rounds; trust rupture, where the trustee kept the entire investment; and trust repair, where the trustee behaved cooperatively following rupture [18]. Trust was operationalized as the amount transferred by the investor to the trustee for investment, averaged across the aggregated rounds of each phase. Liebke, Koppe [18] found no difference between BPD patients and healthy controls (HC) in the average amount invested during the rupture or repair phases but found evidence that patients may transfer less during the cooperative phases, what might loosely be defined as the trust formation phase.

The latter findings were consistent with a 5-round TG study in which feedback on trustee reciprocity was withheld [17]. Results indicated that average levels of trust were lower for patients with BPD than patients with major depressive disorder (MDD) or HC's. Linear trend analyses further revealed that while both control groups progressively increased their investments over the five rounds, growth was not observed for the BPD group. Together these findings suggest that individuals with BPD have a greater propensity towards mistrust when interacting with a new partner, even in the absence of investment loss.

The temporal nature of trust in BPD was also examined in a study focusing on trustee rather than investor behaviour. King-Casas, Sharp [16] used human dyads to elegantly quantify the process of trustee cooperation in negotiating trust rupture and repair by pairing HC investors with either HC or BPD trustees in a 10-round TG. They discovered that in response to declining investments by the investor—a signal of diminishing trust—healthy trustees increased their reciprocity over subsequent rounds, effectively 'coaxing' the investor to send larger investments. In contrast, BPD trustees were more likely to respond by reducing reciprocity further or keeping the entire investment, resulting in increased incidences of breakdown in cooperation with investors sending even less or nothing in subsequent rounds. They also found differential neurological activity in the bilateral anterior insula between BPD and HC trustees following receipt of a smaller investment. This area is associated with defection in social exchanges following norm violations [for reviews see 34, 35], leading King-Casas and colleagues' to propose that atypical social norms may underlie the reduced incidence of coaxing behaviours among BPD trustees, by way of a failure to recognize and/or respond appropriately to the social signals of reduced trust.

Although King-Casas and colleagues' study measured trustee cooperation rather than trust, their findings compliment the BPD trust literature to suggest that individuals with BPD compromise the maintenance of trust by not acting to preserve trust when a rupture has occurred [16]. Considered together, the extant research highlights the anomalous relationship between BPD and interpersonal trust, but also exposes gaps in the research, particularly in understanding how trust dissolves and is rebuilt in relation to BPD. Considering this, the current study

has paired the TG with appropriate methodological and data analytical procedures better suited to address the dynamic and multiphasic nature of trust.

## Measuring change with Discontinuous Growth Modelling (DGM)

In recent years, researchers in the organizational psychology field have advanced the research into trust as a dynamic process by employing longitudinal designs and data analytic approaches suitable to capture how trust unfolds across the various phases of trust [24, 26, 27]. These studies have utilized discontinuous growth modelling [DGM: 36, 37], a derivative of mixed effects modelling that can model longitudinal data whilst accounting for discontinuities in the data such as an experimenter-induced trust violation event. This methodological and analytical pairing has enabled examination into the individual differences and higher-level factors that influence how trust forms, dissolves, and restores.

In addition to allowing researchers to measure changes in trust over time and in response to specific events, the change parameters can be coded to measure either *absolute* change in trust or the *relative* intraindividual fluctuations in trust. While using relative coding is the standard in DGM [e.g., 37, 38], Bliese and Lang recommend that modelling both types of change may better inform theory development and practical applications [36]. For example, how quickly trust is rebuilt in *absolute* terms can be determined by comparing the trajectory of trust during the trust restoration phase to zero, and further test whether this trajectory is modified by the presence of BPD symptoms. On the other hand, how quickly trust is rebuilt *relative* to how quickly it was initially built can be determined by comparing the trajectory of trust during restoration to the trajectory of trust before the violation took place. As a second step, it is then possible to determine whether BPD moderates the *relative* difference in the trajectories. Examining relative change is especially important if there is a notable linear trend during the period before a discontinuity. The comparatively slower rate of trust growth observed in BPD patients in the initial stages of an exchange [17] supports the inclusion of relative coding.

## Current study and aims

The current study uses a multi-round TG and discontinuous growth modelling to examine how BPD impacts interpersonal trust processes, including the development of trust with a new partner, how trust dissolves in response to trust violation, and the restoration of trust in response to trust repair. Based on the methodology adopted in previous works [24, 26, 27], a 15-round TG is used with modified reciprocity to precipitate a distinct trust violation, before resuming a pre-violation rate of reciprocity to precipitate a trust repair. These variations in reciprocity are intended to elicit the three trust phases of formation, dissolution, and restoration. This allows five change parameters to be modelled: three parameters representing the rate of trust growth within each of the phases and two parameters representing the change in the level of trust between the phases.

The current study primarily aims to examine whether the number of BPD traits reported modifies, (a) how much trust decreases immediately following the violation (dissolution transition), and how much trust increases immediately following the repair (restoration transition), and (b) the direction and rate in which trust changes in each phase (formation slope, dissolution slope, and restoration slope). Based on previous findings that BPD is associated with increased mistrust in the early stages of a social exchange [17, 18] and a reduced tendency to utilize trust reparative behaviours to maintain cooperation in a social exchange [16], it is hypothesized that BPD trait count will be associated with a more pronounced decrease in trust after the initial instance of violation, a less pronounced increase in trust after the initial

instance of repair, and a deleterious effect on the trajectories of trust within each of the formation, dissolution, and restoration phases.

Our secondary aims are to examine the fluctuations in trust at an intraindividual level and to determine whether BPD trait count modifies the overall pattern of these fluctuations. In order to determine how the number of BPD traits moderates the rate of trust growth during the dissolution and restoration phases, we switch our focus from *absolute* differences as described in the primary aim, to *relative* differences. Specifically, the study will examine (a) the rate that trust dissolves *relative* to how quickly it was originally built (i.e. dissolution phase vs formation phase); (b) the rate trust is rebuilt *relative* to how quickly it dissolved (i.e. restoration phase vs dissolution phase); and (c) the rate trust is rebuilt *relative* to how quickly it was originally built (i.e. restoration phase vs formation phase).

The uncooperative behaviour observed among the trustees with BPD in King-Casas, Sharp [16] did not appear until the latter half of the TG, suggesting their ability to maintain a trust relationship deteriorated over time. Given that BPD traits are expected to be associated with a slower rate of trust growth during the formation phase, it is hypothesized that even when this initial pattern of trust change is taken into account, trust growth during the latter trust phases will be even slower for those with a high number of BPD traits. In other words, as BPD trait count increases, the trajectory of monetary units transferred during dissolution and/or restoration in comparison to the trajectory of transfers made during formation, will be more negative.

The final hypothesis refers to changes in the rate and direction of change during restoration relative to dissolution. Based on King-Casas et al. [16] finding that trustees with BPD were less likely to coax by increasing reciprocity in order to maintain cooperation, it is possible that in a similar vein, investors with high levels of BPD traits may not respond as favourably to the increase in reciprocity levels in regards to increasing their rate of trust growth. It is hypothesized that even after taking into account the rate of trust dissolving in the dissolution phase, BPD trait count will be associated with a slower rate of trust growth during restoration.

Cognitive reflection—the act of problem solving by engaging in conscious deliberation and suppressing intuitive/impulsive responding [39]—has been positively associated with trust in TG's [40]. Additionally, as BPD is associated with impairments in executive functioning [see 41, 42], altered decision making [see 43], deficits in social problem-solving [see 44], and impulsivity in interpersonal contexts [45], including a measure to control for cognitive reflection may be warranted. In regards to gender, a meta-analysis found that female investors invest significantly less than males in the TG [46]. As BPD is diagnosed at a rate of 3:1 in females compared to males [APA, 47], accounting for the effects of gender when examining the influence of BPD traits on trust is justified. Consequently, in both the absolute and relative models, the main effects of cognitive reflective ability and gender will be controlled.

## Materials and methods

The study was approved by the University of Wollongong ethics committee HE2017/253. All participants provided written informed consent.

### Participant

Participants were undergraduate students from a large Australian university who elected to take part in a psychology research participation program in exchange for course credit. As part of an additional study, after playing the TG, participants were asked to describe their own and their partner's intentions in reference to specific transactions made. Participants whose responses indicated scepticism that their partner was human (and not a computer algorithm)

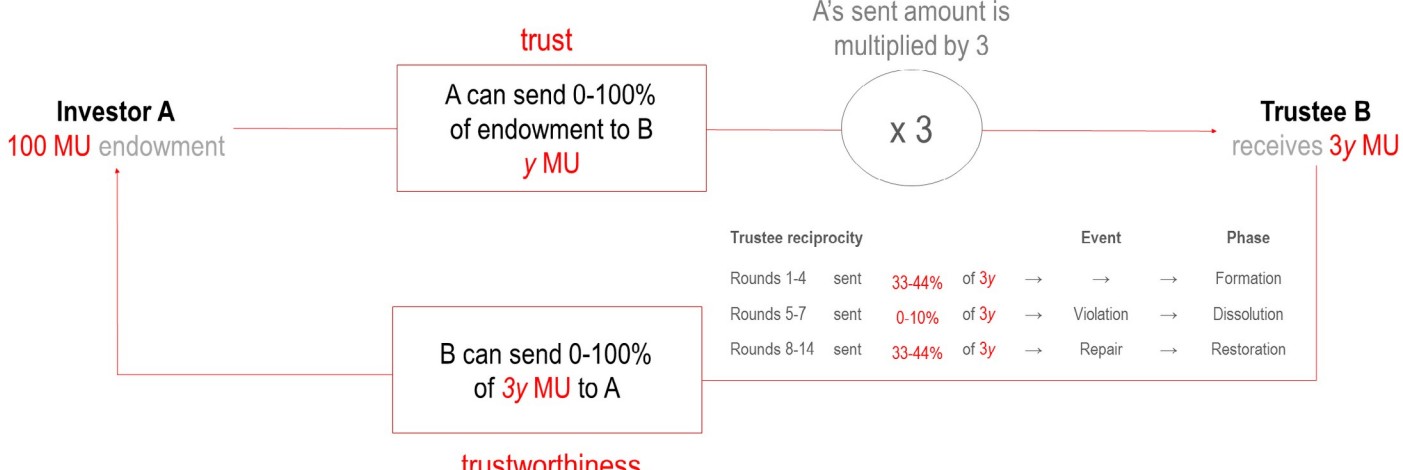

**Fig 1. Trust game structure of a single round.** At the start of each round, investors were allocated 100 monetary units (MUs) from which they could send any amount to the trustee for investment. After receiving the transferred amount multiplied by a factor of 3 (3$y$ MU), the trustee could then return any amount to the investor. Trustee reciprocity was randomized to fall between 33–44% and 0–10% to induce violation and repair, respectively.

were dropped from the final analyses ($n$ = 7), leaving a final sample of 234 (64% female; $M_{age}$ = 20.87, $SD_{age}$ = 5.66). The recruitment information advised participants that the online study was looking at the relationship between economic decision making and various personality variables and would involve questionnaires and playing an investment game.

## Trust Game (TG) protocol

The current study used a multi-round TG comprising of 15 sequential rounds played with the same partner. At the beginning of each round the investor was allocated 100 monetary units (MU) by the experimenter and given the option to send the trustee any proportion from 0–100% for investment. The amount sent was automatically tripled, and the trustee could repay any proportion from 0–100% of the tripled investment to the investor. During rounds 1–4 and 8–14 (inclusive), trustee repayments were randomized to fall between 34–44% of the tripled investment, providing the investor with a return the equivalent of the original investment plus up to 32% profit (range 0–32%). A trustee reciprocity range of 34–44% was selected to ensure repayments would be comparable to the reciprocity ratios observed in human trustees in previous research [16, 46]. During rounds 5–7 (inclusive), trustee repayments were randomized to fall between 0–10% of the tripled investment, providing the investor with a return the equivalent of losing from 70% to 100% of their original investment (due to rounding to the nearest whole number, investments of 1 MU did not incur a loss). This loss was designed to induce a trust violation, and it was repeated over three consecutive rounds to reinforce the participant's perception that the violation was deliberate and not construed as a mistake [27, 48]. Based on this repayment schedule, investments can be divided into three distinct trust phases: formation (rounds 1–5), dissolution (rounds 6–8), and restoration (rounds 9–15). Fig 1 illustrates the process of a single exchange and the rates of trustee reciprocity preceding each event. Following each round, the participant was provided with a summary indicating how much they invested, how much was repaid, and the final sum each party ended up with for that round. As participants saw the summary at the end of each round, trust rates lag trustee reciprocity rates by 1 round.

## Procedure

Participants registered and completed the study online during the study period. After completing a series of questionnaires, participants were given instructions on how to play the TG and were required to correctly answer three test questions to demonstrate their understanding of the game before being permitted to proceed. Despite being told roles would be allocated randomly, participants were all assigned the role of investor. They were also informed that they would be paired with another study participant from a participating academic institution, when in fact the other 'player' was a computer programmed trustee. This form of deception has been used in most of the experiments using the same protocol in a BPD population [e.g., 17, 33]. The use of a computerized agent rather than a human confederate allows standardization of trustee responses in terms of reciprocity levels and response time. Participants were not advised how many rounds they would play as previous research has shown that the defection rate increases when people know when a social exchange will end [49]. Participants were not offered a monetary incentive to participate but at the conclusion of the study, they were automatically allocated course credit.

## Measures

**Trust.** The number of MU that participants entrusted to their partner in each round, ranging from 0 to 100, represents a single behavioural measure of trust. Each participant provided 15 trust measures in total.

**BPD trait count.** The McLean Screening Instrument for Borderline Personality Disorder [MSI-BPD; 50] is a 10-item self-report screening instrument for BPD, with demonstrated internal consistency, validity, test-retest reliability, and in individuals aged 18 to 25, excellent sensitivity (.90) and specificity (.93) for the diagnosis of DSM-IV BPD [APA, 51]. The MSI-BPD has satisfactory internal consistency (Cronbach's $\alpha$ = .74, N = 200) and good test and retest reliability (Spearman's rho = .72) [50]. In the current study the MSI-BPD was used as a measure of BPD trait count and provided a score ranging from 0 to 10.

In our sample the MSI-BPD demonstrated very good internal consistency (Cronbach's $\alpha$ = .84, N = 234). The MSI-BPD additionally had significant moderate positive correlations with the Mental Health Inventory [MHI-5; 52], a measure of general mental health and quality of life (r = .54, p < .001), and the Standardized Assessment of Personality—Abbreviated Scale Self-Report [SAPAS-SR; 53], a measure of general personality psychopathology (r = .54, p < .001). In the current sample, 24% reported 3 or more BPD traits, a level of symptomatology that is considered to reflect the presence of a notable level of borderline pathology [CLPS: 6]. Finally, using the MSI-BPD conservative cut-off of 7/10, 16 participants met caseness for BPD.

**Cognitive reflective ability.** The Cognitive Reflection Test [CRT; 39] is a three-item measure of the willingness to engage in deliberation during a cognitive task. Each item is a deceptively simple mathematical problem in which there is an intuitive but incorrect answer. The CRT aims to measure the ability or disposition to resist responding impulsively by assessing the number of questions answered correctly. Participants are required to answer each question by typing the correct response in digits. Correct responses receive a score of 1 with all other responses scored 0, resulting in a total summed CRT score ranging from 0–3. The CRT has a moderate positive correlation with measures of intelligence and is correlated highly with various measures of decision-making indices [39]. In the current study 48% of participants scored 0, 18% scored 1, 17% scored 2, and 17% scored 3, with an overall sample mean of 1.03, which falls within the range of CRT scores collected from other academic institutions [see 39].

**Post-game trustee appraisals.** After the game participants rated the trustee on fairness (*'Did the other player play fair*?*'*) and trustworthiness (*'Is the other player trustworthy*?*'*) on 5-point Likert-type scales (0 = "Not at all"; 4 = "Absolutely".

**Data analyses.** The NLME package [54], included in the open source software R [55], was used to conduct discontinuous growth modelling (DGM) analyses [see 36, 37]. We tested two-level mixed-effects models, with investment occasions (rounds) at Level 1 nested within individuals at Level 2. Level 1 parameters were coded based on the framework recommended by Singer and Willett [37] and Bliese and Lang [36] to examine changes in the average level of trust between and growth within each of the formation (pre-violation), dissolution, and restoration phases. Five change variables were created to examine *absolute* and *relative* change. According to Bliese and Lang [36], the coding and combination of these change variables into a matrix allows for the regression coefficients to describe the change in the dependent variable in different ways. As we were interested in how individuals respond first to a trust violation and then to a trust restoration, we coded the change variables such that the coefficients reflect the previous stage as the baseline for interpretation. Specifically, the change variables coded for the dissolution phase (DT and DS) use the formation phase as a baseline and change variables coded for the restoration phase (RT and RS) use the dissolution phase as a baseline. The coding and interpretation for each change variable are presented in Table 1a and 1b for absolute and relative change, respectively. This coding allowed for easy interpretation of how individuals responded during the three phases of formation, dissolution, and restoration. Finally, an alternative coding system was used to reflect relative change using the formation phase as a baseline (see Table 1c).

We initially examined Level 1 change by including only Level 1 predictors in a series of models to calculate a basic DGM. Next, we examined the Level 2 model by including BPD trait count as a Level 2 predictor to account for differences in Level 1 change, while also controlling for the main effects of gender and cognitive ability. Snijders and Bosker [56] maintain that as a result of reduced parameter reliability in multilevel analysis, the power to detect cross-level interactions may be insufficient (p. 94). To account for this, a criterion level of $p < .10$ was used for all cross-level interactions effects, and $p < .05$ for all other effects [see also 57, 58], and all tests conducted were two-tailed. We tested a model examining *absolute* change to address our first research aim followed by models examining *relative* change to address our second research aim. For the mixed-effects analyses, all level 2 predictors were *z*-standardized and centered at the sample mean.

## Results

### Descriptive data and intercorrelations

The dataset for the current study can be accessed in S1 Dataset. Table 2 presents the means, standard deviations, and intercorrelations for BPD trait count, MUs transferred within each trust phase, partner appraisals, and cognitive reflection.

BPD trait count had a significant negative correlation with MUs transferred during formation but was not found to be significantly associated with MUs transferred during the dissolution or restoration phases. BPD trait count was not significantly associated with appraisals of trustworthiness or fairness. However, trustworthiness appraisals were positively associated with the number of MUs transferred during all three trust phases while fairness appraisals were positively associated with the amount transferred during the dissolution phase only.

To examine whether there were systematic differences in trustee reciprocity rates we conducted both correlation and ANOVA analyses so that we could treat BPD trait count as a continuous and categorical variable. We created three BPD categories based on number of traits

**Table 1. Coding and interpretation of change variables in the discontinuous mixed-effects growth models[a].**

**a) Absolute change**

| Change variable | Formation | | | | | | | | | | | | | | | Dissolution |
|---|---|---|---|---|---|---|---|---|---|---|---|---|---|---|---|---|
| **Restoration** | | | | | | | | | | | | | | | | **Interpretation of associated parameter estimates in the DGM** |
| Rounds | 1 | 2 | 3 | 4 | 5 | 6 | 7 | 8 | 9 | 10 | 11 | 12 | 13 | 14 | 15 | |
| $TIME_{ti}$ | 0 | 1 | 2 | 3 | 4 | 4 | 4 | 4 | 4 | 4 | 4 | 4 | 4 | 4 | 4 | Linear change of MUs transferred in the formation phase ($\pi_{1i}$) |
| $DT_{ti}$ | 0 | 0 | 0 | 0 | 0 | 1 | 1 | 1 | 1 | 1 | 1 | 1 | 1 | 1 | 1 | Difference in level of MUs transferred immediately following the trust violation ($\pi_{2i}$), Round 6 vs Round 5 |
| $DS_{ti}$ | 0 | 0 | 0 | 0 | 0 | 0 | 1 | 2 | 2 | 2 | 2 | 2 | 2 | 2 | 2 | Linear change of MUs transferred in the dissolution phase |
| $RT_{ti}$ | 0 | 0 | 0 | 0 | 0 | 0 | 0 | 0 | 1 | 1 | 1 | 1 | 1 | 1 | 1 | Difference in level of MUs transferred immediately following the trust repair ($\pi_{4i}$), Round 9 vs Round 8 |
| $RS_{ti}$ | 0 | 0 | 0 | 0 | 0 | 0 | 0 | 0 | 0 | 1 | 2 | 3 | 4 | 5 | 6 | Linear change of MUs transferred in the restoration phase |

**b) Relative change (relative to preceding phase)**

| Change variable | Formation | | | | | | | | | | | | | | | Dissolution |
|---|---|---|---|---|---|---|---|---|---|---|---|---|---|---|---|---|
| **Restoration** | | | | | | | | | | | | | | | | **Interpretation of associated parameter estimates in the DGM** |
| Rounds | 1 | 2 | 3 | 4 | 5 | 6 | 7 | 8 | 9 | 10 | 11 | 12 | 13 | 14 | 15 | |
| $TIME_{ti}$ | 0 | 1 | 2 | 3 | 4 | 5 | 6 | 7 | 8 | 9 | 10 | 11 | 12 | 13 | 14 | Linear change of MUs transferred in the formation phase ($\pi_{1i}$), also referred to as the pre-violation period |
| $DT_{ti}$ | 0 | 0 | 0 | 0 | 0 | 1 | 1 | 1 | 1 | 1 | 1 | 1 | 1 | 1 | 1 | Difference in MUs transferred immediately following the trust violation ($\pi_{2i}$) relative to the predicted transfer amount based on the formation phase (i.e. the expected MUs at Round 6 in the absence of trust violation) |
| $DS_{ti}$ | 0 | 0 | 0 | 0 | 0 | 0 | 1 | 2 | 3 | 4 | 5 | 6 | 7 | 8 | 9 | Linear change of MUs transferred in the dissolution phase relative to the formation phase (i.e. the pre-violation period) ($\pi_{3i}$) |
| $RT_{ti}$ | 0 | 0 | 0 | 0 | 0 | 0 | 0 | 0 | 1 | 1 | 1 | 1 | 1 | 1 | 1 | Difference in MUs transferred immediately following the trust repair ($\pi_{4i}$) relative to the predicted transfer amount based on the dissolution phase (i.e. the expected MUs at Round 9 in the absence of trust repair) |
| $RS_{ti}$ | 0 | 0 | 0 | 0 | 0 | 0 | 0 | 0 | 0 | 1 | 2 | 3 | 4 | 5 | 6 | Linear change of MUs transferred in the restoration phase relative to the dissolution phase (i.e. the pre-repair period) ($\pi_{5i}$) |

**c) Relative change (relative to formation phase)**

| Change variable | Formation | | | | | | | | | | | | | | | Dissolution |
|---|---|---|---|---|---|---|---|---|---|---|---|---|---|---|---|---|
| **Restoration** | | | | | | | | | | | | | | | | **Interpretation of associated parameter estimates in the DGM** |
| Rounds | 1 | 2 | 3 | 4 | 5 | 6 | 7 | 8 | 9 | 10 | 11 | 12 | 13 | 14 | 15 | |
| $TIME_{ti}$ | 0 | 1 | 2 | 3 | 4 | 5 | 6 | 7 | 8 | 9 | 10 | 11 | 12 | 13 | 14 | Linear change in MUs transferred in the formation phase ($\pi_{1i}$), also referred to as the pre-violation period |
| $DT_{ti}$ | 0 | 0 | 0 | 0 | 0 | 1 | 1 | 1 | 0 | 0 | 0 | 0 | 0 | 0 | 0 | Difference in MUs transferred immediately following the trust violation ($\pi_{2i}$) relative to the predicted transfer amount based on the formation phase (i.e. the expected MUs at Round 6 in the absence of trust violation) |
| $DS_{ti}$ | 0 | 0 | 0 | 0 | 0 | 0 | 1 | 2 | 0 | 0 | 0 | 0 | 0 | 0 | 0 | Linear change in MUs transferred in the dissolution phase relative to the formation phase (i.e. the pre-violation period) ($\pi_{3i}$) |
| $RT_{ti}$ | 0 | 0 | 0 | 0 | 0 | 0 | 0 | 0 | 1 | 1 | 1 | 1 | 1 | 1 | 1 | Difference in MUs transferred immediately following the trust repair ($\pi_{4i}$) relative to the predicted transfer amount based on the formation slope (i.e. the expected MUs at Round 9 in the absence of trust violation/repair) |
| $RS_{ti}$ | 0 | 0 | 0 | 0 | 0 | 0 | 0 | 0 | 0 | 1 | 2 | 3 | 4 | 5 | 6 | Linear change in MUs transferred in the restoration phase relative to the formation phase (i.e. the pre-violation period) ($\pi_{5i}$) |

Rounds = measurement occasions in the trust game, $TIME_{ti}$ = linear change, $DT_{ti}$ = dissolution transition, $DS_{ti}$ = dissolution slope, $RT_{ti}$ = restoration transition, $RS_{ti}$ = restoration slope

[a] As this is a complex coding scheme, we asked several experts in the use of DGM to evaluate and ensure the interpretation of the variables based on the matrices presented. See Acknowledgments.

**Table 2. Intercorrelations, means, and standard deviations of study variables.**

|  | 1 | 2 | 3 | 4 | 5 | 6 | 7 |
|---|---|---|---|---|---|---|---|
| 1. BPD | - | | | | | | |
| 2. Formation | -0.18** | - | | | | | |
| 3. Dissolution | -0.01 | 0.34*** | - | | | | |
| 4. Restoration | -0.08 | 0.44*** | 0.55*** | - | | | |
| 5. Trustworthiness | -0.10 | 0.19** | 0.16* | 0.15* | - | | |
| 6. Fairness | -0.10 | 0.04 | 0.15* | 0.07 | 0.49*** | - | |
| 7. Cognitive reflective test | -0.11† | 0.23*** | 0.03 | 0.13† | 0.03 | 0.03 | - |
| Mean | 1.63 | 46.16 | 24.54 | 38.4 | 2.12 | 2.71 | 1.03 |
| SD | 2.34 | 18.95 | 19.14 | 23.29 | 1.08 | 1.18 | 1.15 |

†$p < .10$,

*$p < .05$,

**$p < .01$,

***$p < .001$

$n$ = 234 participants. Spearman correlation.

BPD traits (0–10). Formation/Dissolution/Restoration (0–100). Trustworthiness/Fairness appraisal (1–5). Cognitive reflective test (0–3).

endorsed on the MSI-BPD: Low (0–2), Moderate (3–6), and High (7–10). Due to significant discrepancies in sample size (N = 177/41/16, respectively) and evidence of non-normality and heterogeneity of variance in some distributions, we elected to run a non-parametric ANOVA analysis. Both, the correlation approach and the Kruskal–Wallis H Test did not show any significant difference in trustee reciprocity between BPD categories.

Cognitive functioning had a marginally significant negative correlation with BPD trait count and was positively associated with MUs transferred during formation and dissolution (see Table 1). T-tests suggest that transfers made during formation differ by gender ($p < .01$), with females ($M = 43.75$, $SD = 17.77$) transferring fewer MUs than males ($M = 50.38$, $SD = 20.27$), and during restoration ($p < .001$), with females ($M = 34.39$, $SD = 22.12$) transferring fewer MUs than males ($M = 45.43$, $SD = 23.75$). Females ($M = 23.56$, $SD = 18.42$) and males ($M = 26.24$, $SD = 20.35$) transferred comparable MUs during dissolution ($p$ = ns). T-tests also indicated a marginally significant difference in BPD trait count by gender ($p = .06$), with females ($M = 1.85$, $SD = 2.46$) reporting a slightly higher number of traits than males ($M = 1.25$, $SD = 2.08$).

## Individual differences in trust patterns

We began by testing the random intercept model (null model) to estimate the intraclass correlation coefficient (ICC) to determine how much of the variability in MUs transferred across the 15 rounds resulted from between-person differences. The ICC was .253, indicating 25.3% of the variance in the amount invested across rounds can be explained by properties of the individual. This ICC value is consistent with our expectations based on prior exploration of trust behaviour during the trust game [24] and the knowledge that our experimental design possessed three distinct trust phases.

## Level 1 analyses

Following the procedure established by Bliese and colleagues [36, 59], for each of our absolute and relative analyses, we first generated a linear-only baseline DGM (random intercept model)

**Table 3. Model comparison tests for discontinuous growth models, autocorrelations, and heteroscedasticity.**

| Model | df | AIC | BIC | logLik | Test | L.Ratio |
|---|---|---|---|---|---|---|
| **a) Absolute model** | | | | | | |
| 1. Random Intercept Model | 8 | 33229.79 | 33279.08 | -16606.89 | | |
| 2. Random TIME | 10 | 33206.92 | 33268.54 | -16593.46 | 1 vs 2 | 26.87*** |
| 3. Random TIME & DT | 13 | 33114.75 | 33194.85 | -16544.37 | 2 vs 3 | 98.17*** |
| 4. Random TIME, DT & DS | 17 | 33072.36 | 33177.11 | -16519.18 | 3 vs 4 | 50.39*** |
| 5. Random TIME, DT, DS & RT | 22 | 33057.37 | 33192.92 | -16506.68 | 4 vs 5 | 24.99*** |
| 6. Random TIME, DT, DS, RT & RS | 28 | 32991.98 | 33164.50 | -16467.99 | 5 vs 6 | 77.39*** |
| 7. Autocorrelation Error Structure | 29 | 32968.31 | 33147.00 | -16455.15 | 6 vs 7 | 25.67*** |
| 8. Heteroscedasticity | 29 | 32992.60 | 33171.29 | -16467.30 | 6 vs 8 | 1.38 |
| **b) Relative model (preceding phase as a baseline)** | | | | | | |
| 1. Random Intercept Model | 8 | 33229.79 | 33279.08 | -16606.89 | | |
| 2. Random TIME | 10 | 33113.18 | 33174.80 | -16546.59 | 1 vs 2 | 120.60*** |
| 3. Random TIME & DT | 13 | 33065.67 | 33145.77 | -16519.83 | 2 vs 3 | 53.51*** |
| 4. Random TIME, DT & DS | 17 | 33033.18 | 33137.93 | -16499.59 | 3 vs 4 | 40.49*** |
| 5. Random TIME, DT, DS & RT | 22 | 33002.31 | 33137.87 | -16479.16 | 4 vs 5 | 40.87*** |
| 6. Random TIME, DT, DS, RT & RS | 28 | 32992.08 | 33164.61 | -16468.04 | 5 vs 6 | 22.23*** |
| 7. Autocorrelation Error Structure | 29 | 32968.45 | 33147.14 | -16455.22 | 6 vs 7 | 25.63*** |
| 8. Heteroscedasticity | 29 | 32992.61 | 33171.30 | -16467.31 | 6 vs 8 | 1.47 |
| **c) Relative model (formation phase as a baseline)** | | | | | | |
| 1. Random Intercept Model | 8 | 33229.79 | 33279.08 | -16606.89 | | |
| 2. Random TIME | 10 | 33113.18 | 33174.80 | -16546.59 | 1 vs 2 | 120.60*** |
| 3. Random TIME & DT | 13 | 33103.59 | 33183.70 | -16538.80 | 2 vs 3 | 15.59** |
| 4. Random TIME, DT & DS | 17 | 33109.18 | 33213.93 | -16537.59 | 3 vs 4 | 2.42 |
| 5. Random TIME, DT, DS & RT | 22 | 33037.07 | 33172.63 | -16496.54 | 4 vs 5 | 82.11*** |
| 6. Random TIME, DT, DS, RT & RS | 28 | 32992.06 | 33164.58 | -16468.03 | 5 vs 6 | 57.01*** |
| 7. Autocorrelation Error Structure | 29 | 32968.41 | 33147.10 | -16455.21 | 6 vs 7 | 25.65*** |
| 8. Heteroscedasticity | 29 | 32992.67 | 33171.35 | -16467.33 | 6 vs 8 | 1.39 |

*$p < .05$,

**$p < .01$,

***$p < .001$.

DT = dissolution transition, DS = dissolution slope, RT = restoration transition, RS = restoration slope.

to determine the pattern of change in trust for participants as a whole using the TIME variable.

$$Trust_{ti} = \pi_{0i} + \pi_{1i}TIME_{ti} + \pi_{2i}DT_{ti} + \pi_{3i}DS_{ti} + \pi_{4i}RT_{ti} + \pi_{5i}RS_{ti} + \varepsilon_{ti}$$

This model consisted of an intercept ($\pi_{0i}$), error variance ($\varepsilon_{ti}$), and the change variables: linear change over time ($TIME_{ti}$), dissolution transition ($DT_{ti}$), dissolution slope ($DS_{ti}$), restoration transition ($RT_{ti}$), and restoration slope ($RS_{ti}$), as described in Table 1.

We then compared the random intercept model, which describes the overall trajectory averaged across all participants, with a model that allows the trajectory of each participant to vary. By allowing for random variance across each change coefficient, we can estimate whether there are between-person differences in the pattern of MUs transferred. Beginning with the random intercept model, each consecutive model allows an additional change coefficient to freely vary across participants. Each consecutive model was tested against the previous model using the log-likelihood test (for each model, see rows 1 to 6 of Table 3a–3c below). Results indicate that a model accounting for random effects for all change coefficients was the best

fitting model for both the absolute and relative models, except for the dissolution slope in the model looking at change relative to the formation phase (see Table 3c). However, this does not affect the interpretation of this model since its purpose in the present study is to examine trust changes during the restoration phase in comparison to trust levels and growth before the violation took place.

In the next step, we tested for lag-1 autocorrelation (row 7 of Table 3 for each model) and heteroscedasticity (row 8 of Table 3 for each model). Log-likelihood ratio tests indicated a significantly better fit only when we accounting for autocorrelation. Models that controlled for both error structures simultaneously did not converge. The Final DGM's in Table 4 below provides the parameter estimates and standard errors with random transitions and slopes, and autocorrelation included.

The results of the absolute model indicate that, on average, the MUs transferred decreased significantly immediately following the trust violation ($est_{DT} = -20.60$, $p < .001$) and increased significantly immediately following the trust repair ($est_{RT} = 10.83$, $p < .001$). This confirms that the experimental manipulation of trustee reciprocity to create trust violation and trust restoration was successful. There was no evidence of a linear trend in MUs transferred during the formation ($est_{FS} = -.12$, $n.s.$) and dissolution phases ($est_{DS} = -.80$, $n.s.$), but there was a positive linear trend in MUs transferred during the restoration phase ($est_{RS} = 1.26$, $p < .01$). The results of the corresponding relative model further indicate that the immediate shift in MUs transferred following the trust violation ($est_{DT} = -20.49$, $p < .001$) resulted in transfer amounts that were significantly lower than what would be expected if the violation had not occurred. Furthermore, the immediate shift in the MUs transferred following the trust repair ($est_{RT} = 11.63$, $p < .001$) resulted in transfer amounts that were significantly higher than what would be expected if the repair had not occurred. However, when using the formation phase as a baseline, this increase in transfer amounts at repair was still significantly lower than what would be expected if neither the violation nor repair had occurred ($est_{RT} = -10.91$, $p < .01$). Regarding trust growth, the slope for the dissolution phase was not found to differ significantly from the slope during the formation phase ($est_{DS} = -.69$, $n.s.$). However, the positive linear trend in MUs transferred during restoration was steeper than that observed during the formation phase ($est_{RS} = 1.38$, $p < .10$) and during the dissolution phase ($est_{RS} = 2.07$, $p < .10$).

## Level 2 analyses

To test for systematic differences in MUs transferred between individuals due to differences in the number of BPD traits reported, BPD trait count was added as a Level 2 predictor for each

**Table 4. Final discontinuous growth models.**

| Variables | Absolute | | Relative (to preceding phase) | | Relative (to formation phase) | |
|---|---|---|---|---|---|---|
| | Est | SE | Est | SE | Est | SE |
| Intercept | 46.42*** | 1.51 | 46.42*** | 1.51 | 46.42*** | 1.51 |
| Formation slope (FS) | -0.12 | 0.58 | -0.12 | 0.58 | -0.12 | 0.58 |
| Dissolution transition (DT) | -20.60*** | 2.39 | -20.49*** | 2.79 | -20.49*** | 2.79 |
| Dissolution slope (DS) | -0.80 | 1.16 | -0.69 | 1.28 | -0.69 | 1.28 |
| Restoration transition (RT) | 10.83*** | 1.87 | 11.63** | 2.63 | -10.91** | 4.15 |
| Restoration slope (RS) | 1.26** | 0.40 | 2.07† | 1.25 | 1.38† | 0.71 |

†$p < .10$,

*$p < .05$,

**$p < .01$,

***$p < .001$, tests are two-tailed, $n$ = 234 participants, 3510 observations.

**Table 5.  Discontinuous mixed-effects growth models predicting trust as a function of BPD trait count after controlling for gender and cognitive reflective ability.**

|  | Absolute | | Relative (preceding phase) | | Relative (formation phase) | |
|---|---|---|---|---|---|---|
|  | Est | SE | Est | SE | Est | SE |
| Intercept | 49.14*** | 2.02 | 49.12*** | 2.02 | 49.20*** | 2.02 |
| Formation slope (FS) | -0.08 | 0.57 | -0.08 | 0.57 | -0.08 | 0.57 |
| Dissolution transition (DT) | -20.74*** | 2.40 | -20.67*** | 2.79 | -20.67*** | 2.80 |
| Dissolution slope (DS) | -0.76 | 1.16 | -0.69* | 1.27 | -0.69 | 1.27 |
| Restoration transition (RT) | 10.55*** | 1.86 | 11.31** | 2.63 | -11.42** | 4.12 |
| Restoration slope (RS) | 1.29** | 0.40 | 2.05 | 1.26 | 1.36 | 0.71 |
| Gender (Female) | -4.30* | 2.15 | -4.28* | 2.15 | -4.41* | 2.15 |
| Cognitive reflective ability | 2.34* | 1.04 | 2.34* | 1.04 | 2.36* | 1.03 |
| BPD | 0.17 | 1.49 | 0.17 | 1.49 | 0.18 | 1.50 |
| FS * BPD | -1.12† | 0.57 | -1.12† | 0.57 | -1.12† | 0.57 |
| DT * BPD | 2.13 | 2.40 | 3.25 | 2.79 | 3.25 | 2.79 |
| DS * BPD | 2.33* | 1.16 | 3.45** | 1.27 | 3.45* | 1.27 |
| RT * BPD | -3.83* | 1.85 | -6.16* | 2.62 | 7.43† | 4.11 |
| RS * BPD | 0.43 | 0.40 | -1.90 | 1.25 | 1.55* | 0.71 |

†$p < .10$,

*$p < .05$,

**$p < .01$,

***$p < .001$, tests are two-tailed, $n$ = 234 participants, 3510 observations. BPD = Number of BPD traits reported on MSI-BPD. BPD and cognitive reflective ability were $z$-standardized and centered at the sample mean.

of the Level 1 components. The associated Level 2 equations are as follows:

$$\pi_{0i} = \beta_{00} + \beta_{01}(BPD)_i + r_{0i}$$

$$\vdots$$

$$\pi_{5i} = \beta_{50} + \beta_{51}(BPD)_i + r_{5i}$$

Based on results from separate main effects models for gender (est$_{\text{Female}}$ = -5.64, $p < .01$) and cognitive reflective ability (est$_{\text{Cognitive}}$ = 2.52, $p < .01$) which found that being female or having low cognitive reflective ability were associated with smaller transfers overall, we controlled for both in our final interaction models. Results for the final discontinuous mixed-effects models are presented in Table 5.

Fig 2 graphs the effects of BPD on the overall change pattern of MUs transferred. MUs transferred at each measurement occasion was predicted for individuals with a high (1 SD above the sample mean) and low (1 SD below the sample mean) BPD trait count, contrasted with predicted MUs transferred for individuals scoring at the sample mean of BPD. Results for the absolute model are reported first, followed by an outline of any notable deviations observed when looking at relative change.

Results do not indicate a main effect of BPD trait count. With respect to changes in MUs transferred, it was hypothesized that BPD trait count would have a deleterious effect on the amount transferred at each transition, representing acute changes in trust in reaction to violation and repair, and on the rate of change of MUs transferred during each phase, representing the rate of trust growth or decline.

**Formation slope.**   We predicted that BPD would be associated with a slower rate of trust growth during the formation phase. As indicated by Table 5, this hypothesis was confirmed for

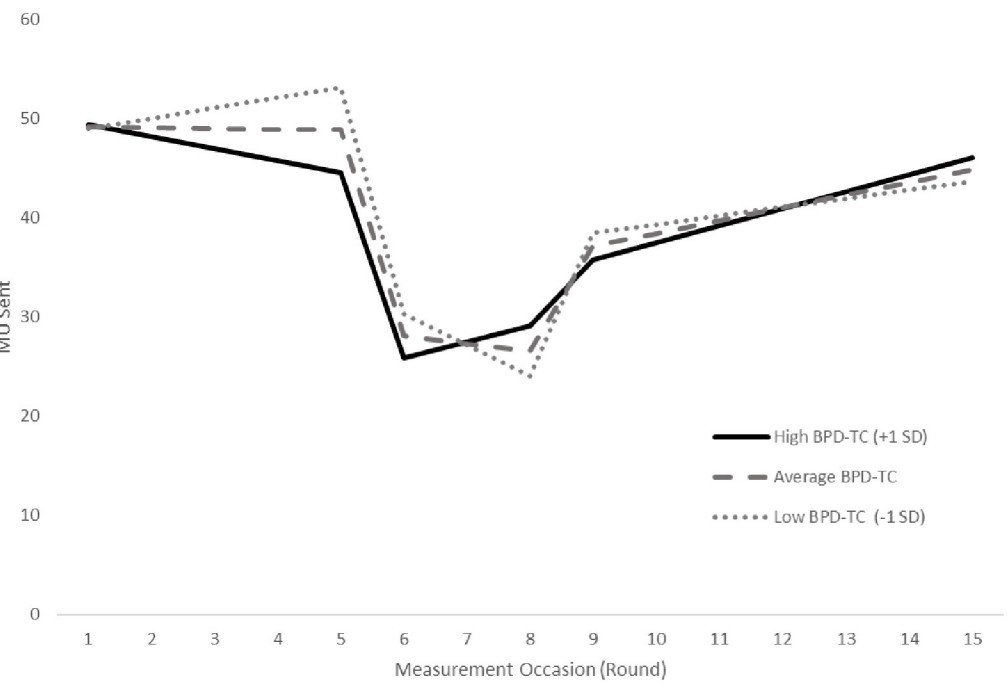

**Fig 2. Number of MUs sent by the investor as a function of Borderline Personality Disorder Trait Count (BPD-TC).** High BPD-TC are investors who scored one or more standard deviations above the mean number of BPD traits on the McLean Screening Instrument for Borderline Personality Disorder, and Low BPD-TC are investors who scored one or more standard deviations below the mean number of BPD traits.

the relationship between BPD trait count and the trajectory of trust during the formation phase where we found a significant negative linear trend ($est_{FS*BPD}$ = -1.12, $p < .10$). This result indicates that individuals with a higher number of BPD traits have a more pronounced decline in MUs transferred during the trust formation phase. This was additionally confirmed in a model which isolated only the first 5 rounds of the game to measure trust change ($est_{FS*BPD}$ = -1.02, $p < 0.10$).

**Dissolution transition.** It was predicted that BPD would be associated with a larger immediate decrease in the level of trust immediately after the trust violation. The hypothesis for the relationship between BPD trait count and the dissolution transition was not supported. While there was a decrease in MUs transferred following the first exposure to the trust violation ($est_{DT}$ = -20.74, $p < .001$), the size of the decrease appeared to be similar regardless of BPD trait count ($est_{DT*BPD}$ = 2.13, *n.s.*).

**Dissolution slope.** It was hypothesized that BPD would be associated with a slower rate of trust growth during the dissolution phase. As indicated by Table 5, while BPD trait count appeared to be a substantive moderator on the change in MUs transferred during the dissolution phase, the effect was not in the hypothesized direction. Rather, there was a significant positive linear trend between BPD trait count and the amount transferred during dissolution ($est_{DS*BPD}$ = 2.33, $p < .05$). This result indicates that individuals with a higher number of BPD traits have a more pronounced increase in the rate of MUs transferred during the dissolution phase, rather than a decline as predicted.

The comparative relative model is used to understand the impact of the trust violation on the intra-individual patterns of trust. It was hypothesized that BPD would be associated with a slower rate of trust growth during the dissolution phase relative to the trust growth observed in the formation phase. Once again, BPD was significantly associated with the rate at which

trust changed during dissolution relative to formation, but not in the hypothesized direction (est$_{DS*BPD}$ = 3.45, $p$ < .01). In relative terms, BPD was associated with a rate of growth in MUs transferred during dissolution that was greater than the rate of growth during formation.

**Restoration transition.**    It was predicted that BPD would be associated with a smaller increase in the level of trust immediately following the trust repair. As hypothesized, BPD trait count appeared to be a substantive moderator on the increase in the number of MUs transferred at the restoration transition (est$_{RT*BPD}$ = -3.83, $p$ < .05). High BPD trait individuals had a significantly less pronounced increase in MUs transferred following the first instance of repair. This effect was even more pronounced when considering the shift in trust levels following repair relative to the level of trust predicted had the repair never occurred (est$_{RT*BPD}$ = -6.16, $p$ < .05). In other words, for persons high on BPD trait count, trust levels increased immediately following restoration but to a significantly less extent than persons low on BPD trait count. Finally, when considering the shift in trust levels following repair relative to the formation phase, BPD was associated with a larger number of units transferred (est$_{RT*BPD}$ = 7.43, $p$ < .10). In other words, the level of trust (at round 9) was greater than what would have been expected based on the formation phase pattern of trust change.

**Restoration slope.**   It was predicted that BPD would be associated with a slower rate of trust growth during the restoration phase. Our hypothesis for the relationship between BPD trait count and the restoration slope was not supported. The linear pattern in MUs transferred during the restoration phase (est$_{RS}$ = 1.29, $p$ < .01) was at a similar rate regardless of BPD trait count (est$_{RS*BPD}$ = .43, $n.s.$).

With reference to relative change, it was predicted that BPD would be associated with a less pronounced rate of trust growth during the restoration phase relative to the formation phase. However, while BPD trait count was found to significantly moderate the rate of relative change during restoration, it was not in the hypothesized direction. BPD trait count was associated with a more pronounced positive linear trend in the number of MUs transferred during restoration relative to the linear trend in transfers observed in the formation phase (est$_{RS*BPD}$ = 1.55, $p$ < .05). BPD was associated with a rate of growth in MUs transferred during restoration that was greater than the rate of growth during formation.

It was also predicted that BPD would be associated with a less pronounced rate of trust growth during the restoration phase even after taking into account the trajectory during dissolution. Results indicate a non-significant coefficient trending in the hypothesized direction (est$_{RS*BPD}$ = -1.90, $p$ = .13). It should be noted however that we expected BPD trait count to be associated with faster dissolving trust in response to the violation, but we found the opposite; indeed, trust grew at a faster rate. In light of this, the non-significant trend described above suggests that, in comparison to the faster growth observed in the dissolution phase, trust growth was tempered during restoration at higher BPD trait counts.

**Trust change as a function of BPD caseness.**    To support the clinical utility of our findings the level 2 analyses were repeated using the MSI-BPD recommended cutoff values to group participants into likelihood of "caseness" ($N$ = 16) and "noncaseness" ($N$ = 218). Results reported are limited to key findings, but a complete account can be found in the supporting files (S1 Table).

In contrast to the BPD trait model, BPD caseness was not found to modify the rate of trust growth during the formation phase, although there were non-significant trends in the expected direction (est$_{FS*caseness}$ = -3.57, $p$ = .12). BPD caseness was found to positively moderate trust growth during the dissolution phase, both relative to the formation phase (est$_{DS*caseness}$ = 14.32, $p$ < .01) and in absolute terms (est$_{DS*caseness}$ = 10.74, $p$ < .05). In response to the repair, BPD caseness was associated with a smaller absolute increase in MUs transferred (est$_{RT*caseness}$ = -14.08, $p$ < .10). For those meeting BPD caseness, trust levels were also lower after the first

instance of repair than that predicted by the trajectory of growth during the dissolution phase (est$_{RT*caseness}$ = -8.29, $p$ < .10). However, caseness was not found to modify the level of trust following repair relative to the level expected based on the formation trajectory (est$_{RT*caseness}$ = 21.32, $p$ = .19). Finally, the rate of trust growth during restoration for those meeting BPD caseness was faster than trust growth during formation (est$_{RS*caseness}$ = 6.02, $p$ < .05), and slower than trust growth during dissolution (est$_{RS*caseness}$ = -8.29, $p$ < .01).

## Discussion

The present study used discontinuous growth modelling with a trust game to investigate the relationship between BPD and trust behaviours. It examined how BPD trait count modified the level and trajectory of trust as it formed with a new partner, dissolved in response to trust violation, and was rebuilt in response to trust repair. The results suggest that trust behaviour in individuals with a high number of BPD traits may be broadly classified as cautious and mistrustful in the beginning of a new relationship, even when the other party is behaving in a cooperative and trustworthy manner, and conversely, during and after a trust violation, trust appears to grow markedly, even in the face of repeated betrayals.

When interacting with a new and cooperative partner, high BPD trait individuals became progressively less trusting. That is, despite earning up to 32% profit on each investment made during this period, as the number of BPD traits increased, individuals progressively reduced the number of MUs transferred. A trend in this direction was also observed in those whose trait levels indicated likely caseness for BPD. This result augments previous findings indicating that average trust levels and trust growth are lower for people with BPD when playing the TG with a new and not uncooperative partner [17, 18]. Regarding the impact of BPD traits on trust behaviour following violation and repair, the findings highlight the benefits of using a method of analysis that is responsive to the dynamic, multiphasic nature of trust. While previous research found average trust levels during dissolution and restoration to be comparable between BPD patients and HC's [18], the current study found BPD had a paradoxical influence on trust patterns after violation and repair. If increasing mistrust defines the pre-violation trust patterns of high BPD trait count individuals, incessant trust growth would best characterise the violation and post-violation periods.

First, despite experiencing three consecutive rounds in which the trustee kept all or most of the investment, BPD trait count was associated with making progressively larger transfers, a finding contrary to what we expected based on King-Casas and colleagues observation that trustees with BPD were less likely to respond to diminishing cooperation via increasing reciprocity [16]. Given that the immediate reduction in the amount transferred following the first instance of violation was of a comparable magnitude across all levels of BPD symptomatology, even when the pre-violation trajectories were accounted for, it is likely that the three violation rounds were recognized as a norm violation irrespective of level of borderline pathology. This is consistent with findings that individuals with BPD can accurately appraise the fairness of trustee reciprocity [33]. It is possible that the decreases in investment size signalling diminishing trust were more nuanced in the King-Casas study [16], whereas our trust violation was unambiguous and therefore, may have prompted an unconventional response. The disparate coaxing behaviours between King-Casas' trustees and our investors may also reflect a property of the roles. It is possible that the powerlessness inherent in the trustee role—it is the investor who decides whether to initially engage—elicits a more aggressive response in people with BPD, while as investor, they can choose whether to be benevolent knowing that they can stop engaging at any point.

Regarding trust behaviours following repair, the size of the increase in MUs transferred following the initiation of trust repair became less pronounced as trait count increased. For persons who reported 9 or 10 BPD traits, there was a decrease rather than the expected increase. In other words, those with the highest levels of BPD symptoms paradoxically reduced the size of their next transfer in response to the trustee's return to cooperative play. It is possible that the first sign of reparative action by the trustee elicited caution and suspicion in higher trait individuals. In fact, if a prediction were to be made about how many MUs high BPD individuals would have sent at that point, had the violation continued, it would be a markedly higher amount than that sent in response to the repair. On the other hand, perhaps individuals with a higher BPD trait count had sought to coax higher returns from the trustee, and upon achieving this objective, stopped the coaxing behaviour and reduced their investments. Finally, it was found that that investors, in general, were on average observed to invest progressively larger amounts throughout the restoration phase. When this growth was considered relative to the growth observed in the formation phase, BPD traits were associated with a comparatively faster rate of growth. That is, trust in high BPD trait persons was observed to restore at a more accelerated rate than it was formed in the period before the violation took place. In fact, as trait count increased, trust grew at a faster rate during both dissolution and restoration than it had during formation.

This pattern of intraindividual fluctuations does not appear to map on to the intraindividual trust fluctuations of the general population [24, 26], with trust during each phase largely appearing to flow in the opposite direction. This suggests that although the ability to recognize norm violations does not appear to be compromised, BPD is associated with intraindividual changes in *trust behaviours* that are socially atypical. This has serious implications for individuals with BPD in terms of how they may be experienced by others during interpersonal interactions. Social cognition processes are believed to be engaged when individuals make strategic interpersonal decisions [60–62]. To maintain a mutually beneficial equilibrium in the TG, each party must be able to recognize, decipher, and respond appropriately to the signals sent. In the case of repeated interactions with the same partner such as in an iterated TG, the intraindividual fluctuations in behaviour communicate meaningful interpersonally relevant data.

In addition to being able to model the mind of the other, Kishida and colleagues [63] also propose that fair social exchange requires three computational capacities in each agent: to compute the social norms associated with such an exchange, to recognize deviations from said norms, and to respond appropriately considering these deviations. In other words, each agent must not only recognize and ascribe meaning to the intraindividual fluctuations of social signals emitted by their partner, but also determine if these fluctuations are socially normative. For example, previous research using a college student population found that the normative pattern in intraindividual fluctuations is that the *restoring* of trust appears to be a lengthier process than forming trust with a new partner [24]. The socially non-normative pattern of intraindividual fluctuations associated with BPD symptomatology may be perceived by others as incomprehensible, unexpected, and perhaps even odd interpersonal dynamics. In as few as 15 rounds of social exchange, the current study showed that borderline pathology predicts a paradoxical relational style that may invite confusion, whereby betrayal begets trust and cooperation, mistrust.

It is conceivable that these kinds of behaviours and preferences are likely to compromise the development of healthy relationships and may lead to relationship breakdowns or attract partners who may perpetuate these potentially deleterious relational dynamics [64]. Our results also support previous research in which individuals with BPD have been found to demonstrate greater acceptance of and perhaps a preference for inequitable treatment. For

instance, findings from economic game studies suggest that compared to HC's, BPD individuals are more likely to accept unfair offers [65] and reject fair offers [66], and express a greater preference for an unfair interaction partner, and lower preference for a fair interaction partner [67]. While much of the trust literature in borderline populations has focused on BPD tendencies towards mistrust and lack of cooperation, our analytical methodology has highlighted the other extreme, the tendency to engage in trusting behaviour in contexts warranting prudence. Indeed, findings from a social network analysis study has showed that BPD is associated with reduced discrimination in differentiating whom in their social network they seek advice and emotional support from [8]. That is, despite the tendency towards trust-compromising beliefs, appraisals, and behaviours, people with BPD may also trust more haphazardly or arbitrarily.

## Limitations & future directions

The current study has several limitations. First, a clinical sample was not used. Previous research has primarily used patients with a BPD diagnosis, so it is possible our findings may not apply to individuals with severe and persistent BPD impairments. It is important to note however, that interpersonal disturbances in nonclinical samples of people with borderline traits are almost as profound as in clinical samples [68], and that young adults with sub-clinical borderline features are more likely to exhibit interpersonal dysfunction at a two-year follow up than their healthy counterparts [69]. Second, we used a simulated trustee rather than an actual human being, and therefore created interactions where the trustee was unconditionally cooperative or uncooperative, regardless of our participants' behaviour. While standardized trustee reciprocity was chosen intentionally to create the distinct phases of trust, doing so limits the conclusions that can be drawn since interpersonal trust is a dyadic process [70]. Third, participants were not offered financial incentives to participate. By not tying the game results to a financial reward, participants may have been less motivated to take the game seriously. Fourth, BPD is most typically associated with relational disturbances in close relationships. It is not clear whether the behaviour observed in a low stakes game with an anonymous partner would reflect trust behaviours in close personal relationships. For example, Miano et al. [71] found that within the context of intimate relationships, individuals with BPD compared to HC's appraised their romantic partner as less trustworthy after discussing a relationship threatening or personally threatening topic, whereas appraisal ratings were comparable following discussion of a neutral topic. It is therefore likely that the experience of a trust violation and repair within the context of an intimate relationship may evoke a more marked or varied response.

Our findings and methodology provide a solid foundation upon which researchers can examine the nuanced factors and processes that may underlie these incongruous trust dynamics, and therefore help inform more targeted interventions. For example, previous research has looked at the effect of underlying attachment insecurity on trust behaviours [e.g., 72–74]. Our methodology could be used to elicit potentially attachment salient events such as a trust violation, and examine whether attachment insecurity underpins the effect of borderline pathology on trust behaviours. Our findings should also be replicated using a patient sample to allow clinical inferences to be made.

## Conclusions

To the best of our knowledge, this is the first study with a BPD focus to use discontinuous growth modelling with a trust game to examine trust as a dynamic and multiphasic process. Specifically, the study revealed the trajectories of trust as it forms, dissolves, and restores in response to trust violation and repair. It also explained how these trajectories varied as a function of the number of BPD traits reported. Showing how trust fluctuates within the individual

and how symptom count modifies the magnitude and direction of the fluxes, provides a richer and more nuanced understanding of how people with BPD traits engage with others in trust-altering contexts. This approach uncovered a paradoxical style of social exchange where social norms appear to be contradicted, thereby creating interpersonal encounters that are seemingly ambivalent, aberrant, and puzzling. By adopting a design and analytical methodology that recognizes the dynamic nature of trust, the study uniquely illustrated how relational disturbances may be produced and maintained in a BPD population.

## Supporting information

**S1 Dataset. Dataset for study.**
(CSV)

**S1 Table. Discontinuous mixed-effects growth models predicting trust as a function of BPD caseness after controlling for gender and cognitive reflective ability.**
(DOCX)

## Acknowledgments

The authors would like to thank Professors Paul Bliese and Jonas Lang for statistical consultation, and Yossef Abramov for developing the online trust game platform.

## Author Contributions

**Conceptualization:** Gamze Abramov.

**Data curation:** Gamze Abramov.

**Formal analysis:** Gamze Abramov, Sebastien Miellet, Jason Kautz.

**Investigation:** Gamze Abramov.

**Methodology:** Gamze Abramov.

**Project administration:** Gamze Abramov.

**Resources:** Gamze Abramov.

**Supervision:** Sebastien Miellet, Brin F. S. Grenyer, Frank P. Deane.

**Visualization:** Gamze Abramov.

**Writing – original draft:** Gamze Abramov.

**Writing – review & editing:** Gamze Abramov, Sebastien Miellet, Jason Kautz, Brin F. S. Grenyer, Frank P. Deane.

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
