## [Decision Letter · Decision Letter 0]

1 Jun 2020

PONE-D-20-05024

The paradoxical decline and growth of trust as a function of borderline personality disorder trait count: Using discontinuous growth modelling to examine trust dynamics in response to violation and repair.

PLOS ONE

Dear Dr. Abramov,

Thank you for submitting your manuscript to PLOS ONE. After careful consideration, we feel that it has merit but does not fully meet PLOS ONE’s publication criteria as it currently stands. Therefore, we invite you to submit a revised version of the manuscript that addresses the points raised during the review process.

Actually, the paper requires only very minor revisions. Thus, please follow the suggestions made by the reviewers so that we can proceed to publication soon.

We look forward to receiving your revised manuscript.

Kind regards,

Christiane Schwieren, Dr.

Academic Editor

PLOS ONE

Journal Requirements:

'This research was supported by Higher Degree Research funding from the University of Wollongong.'

'The authors received no specific funding for this work.'

Reviewers' comments:

Reviewer's Responses to Questions

**Comments to the Author**

1. Is the manuscript technically sound, and do the data support the conclusions?

The manuscript must describe a technically sound piece of scientific research with data that supports the conclusions. Experiments must have been conducted rigorously, with appropriate controls, replication, and  ample sizes. The conclusions must be drawn appropriately based on the data presented. 

Reviewer #1: Yes

Reviewer #2: Yes

2. Has the statistical analysis been performed appropriately and rigorously? 

Reviewer #1: Yes

Reviewer #2: Yes

3. Have the authors made all data underlying the findings in their manuscript fully available?

Reviewer #1: Yes

Reviewer #2: Yes

4. Is the manuscript presented in an intelligible fashion and written in standard English?

Reviewer #1: Yes

Reviewer #2: Yes

5. Review Comments to the Author

Reviewer #1: I found this article interesting, compelling, and a provocative addition to the literature on trust in BPD. The authors used a 15 round trust game with variations in the amount that trustees returned during different phases of the game. Paradoxically, BPD severity was associated with declining trust when interacting with a new and cooperative partner, but increasing trust after multiple instances of trust violation by that partner. Higher BPD severity also predicted fast restoration of trust than originally formed. It is difficult to understand why a person would respond in this way, mistrusting benevolence and trusting malevolence, which increases the appeal of the study. The study is supported by a thoughtful research design which is further enhanced by the use a sophisticated statistical approach. The study limitations have been well supported. I would like to see this study replicated with a clinical sample.

I read the paper several times, but only found minor critiques:

1. I don’t really see the need for the BPD categorical analyses, particularly given the small number of cases. I don’t mind it’s inclusion, but particularly given these are not clinician diagnoses, the dimensional approach seems superior.

2. I wonder if the findings could be interpreted through an interpersonal/attachment lens. People with BPD tend to oscillate in terms of interpersonal and attachment style, withdrawing when connection is available and approach when it is not. Once the partner returned to cooperative play, people with higher BPD again withdraw. This mirrors the tendency of patient with BPD to show severe mistrust of clinicians and at the same time significant trust towards sometimes exploitative and dangerous romantic partners.

3. To align with previous literature, I have most typically seen the player who invests money in the trustee called the 'investor' rather than the 'truster'.

Reviewer #2: It was a pleasure to read this manuscript and I really appreciate the approach the authors chose to investigate trust in interpersonal exchanges. The only aspect that needs to be clarified from my point of view relates to the repay ratios during formation and restoration of trust. It seems that a ratio of 33-44% is rather low, since with 33% the trustor would receive only the amount he/she invested before. One may ask whether this can really be regarded as a fair behaviour of the trustee. It seems that the repay ratios were randomly varied across participant. Please provide more information on how this was done (is it possible that single participants received in all trials only 33% repay ?). Can the authors make sure that the repay ratios were comparable for participants in the three BPD groups in general, but also during the course of the different trials during formation and restoration ? Please provide the corresponding data.

6. PLOS authors have the option to publish the peer review history of their article (what does this mean?). If published, this will include your full peer review and any attached files.

Reviewer #1: No

Reviewer #2: Yes: Stefanie Lis

---

## [Author Response · Author response to Decision Letter 0]

29 Jun 2020

As our responses contain tables, please see the attached 'Response to Reviewers' document.

---

## [Editor Report · Decision Letter 1]

1 Jul 2020

The paradoxical decline and growth of trust as a function of borderline personality disorder trait count: Using discontinuous growth modelling to examine trust dynamics in response to violation and repair.

PONE-D-20-05024R1

Dear Dr. Abramov,

We’re pleased to inform you that your manuscript has been judged scientifically suitable for publication and will be formally accepted for publication once it meets all outstanding technical requirements.

Kind regards,

Christiane Schwieren, Dr.

Academic Editor

PLOS ONE
---

## [Editor Report · Acceptance letter]

8 Jul 2020

PONE-D-20-05024R1 

The paradoxical decline and growth of trust as a function of borderline personality disorder trait count: Using discontinuous growth modelling to examine trust dynamics in response to violation and repair. 

Dear Dr. Abramov:

I'm pleased to inform you that your manuscript has been deemed suitable for publication in PLOS ONE. Congratulations! Your manuscript is now with our production department. 

Kind regards, 

on behalf of

Dr. Christiane Schwieren 

Academic Editor

PLOS ONE